## OPINION

### Generative AI in academia: Efficiency *versus* scholarship

Daniela Schnitzler

*Institute of Neuroscience and Biopsychology for Clinical Application, Medical School Berlin, Berlin, Germany*

Email: daniela.schnitzler@medicalschool-berlin.de

Handling Editors: Kim Barrett & Vaughan Macefield

The peer review history is available in the Supporting Information section of this article (https://doi.org/10.1113/JP290275#support-information-section).

### Introduction

For myself and many others the increasing prevalence of commercially available generative artificial intelligence (GenAI) tools has called into question the fundamental nature of academic work. In many cases this has sparked debate, polarising those enthusiastically in favour of GenAI use *versus* those with critical and sceptical perspectives (Guest et al. 2026; Pisica et al. 2023). In this opinion piece I aim to distil my thoughts around the use of GenAI in academia from the perspective of an early-career researcher.

Generally there is no argument that the *appropriate* implementation of GenAI can be useful to reduce the trivial and mundane work that slows us down, much like other tools in our modern arsenal such as search engines or autocorrect. Nonetheless, the apparent overzealous use of GenAI by some, in an attempt to alleviate the responsibilities of academic work, is deeply concerning. In fact given the potential harm that may be caused by GenAI, it is therefore unsurprising that universities are currently debating and creating new policies to contend with this burgeoning threat to teaching, research and knowledge production overall.

Indeed, the ethical implications of GenAI use in academia have been discussed elsewhere, for example in medical scientific research (Resnik & Hosseini 2025) or academic writing (Cheng et al. 2025; Dolunay & Temel 2024; Miao et al. 2024). These highlight how the nature of GenAI may skirt ethical academic conduct.

However, beyond the ethical perspectives this writing hopes to inspire introspection through a reflection on the purposes of academic research, learning, teaching and scholarship in turn, from the point of view of an early-career researcher pursuing an academic career through the uncharacteristically rapid changes facing the academic landscape. Here I suggest that many of the facets intrinsic to academia cannot simply be replaced or expedited through GenAI, and to do so is antithetical to academic thought and intellectual pursuits.

### Research

As academic researchers we are extremely fortunate to have careers that allow us to challenge our intellects and construct a deeper understanding of complex ideas. Plainly our jobs as researchers boil down to planning ways to investigate the world, creatively solving problems and considering all paths the evidence might lead us to, as well as the meaning and consequences of each trajectory. Further simplified, regardless of technical or methodological approaches, agnostic to subject or discipline, our jobs are to think.

However, the enthusiasm for adopting new GenAI-based strategies undermines many of the intellectual pursuits intrinsic to academic research. Apart from its uses for purely computational tasks, such as data sorting, it seems many are eager to 'outsource thinking' to GenAI, obfuscated by claims of increasing productivity, efficiency and output. For example, instead of critical thought and deliberate engagement with a paper, GenAI can be used to summarise its findings and generate content to directly adopt into a new piece of work (e.g. Elicit; SciSpace). However the absence of critical judgement of the content, quality of the work or wider meaning of the result render this AI-generated output functionally worthless to the academic researcher, the reader and the foundations of research itself.

These foundations of research depend entirely on a desire for continuous scholarship and intellectual growth, which are weakened by substituting human knowledge production for GenAI output. Using our judgement

and experience, we absorb new information from varied sources, both broad and narrow, thereby conceiving novel ideas based on engagement with heterogeneous perspectives and voices. However, with GenAI the margins of diversity are much narrower, the outputs more discrete, and the opportunities to truly absorb and process new information are stifled (Cheng et al., 2025; Liao et al., 2023; Kim, 2024). With GenAI probabilistically returning the most dominant ideas and perspectives, we cannot appreciate the diversity of thought that the global academic community has to offer.

Furthermore, GenAI is a tool that semi-convincingly creates content based on existing information in an approximation of human writing, yet it cannot reason the way a researcher can (Liao et al., 2023). However, moving beyond any potential technical limitations of GenAI, even if this tool had the capacity to flawlessly mimic the creative processes of human thought and reason, the question remains, why any researcher would want to outsource tasks that are not only integral to scholarship but also underlie the ideation of novel and creative concepts, and ultimately define our roles as scholars and thinkers. Arguably our unique, human abilities to think laterally, abstractly and creatively are what drive innovation and progression. Being able to engage these abilities on a daily basis is a testament to our ambition, perseverance and fortitude to pursue our careers as academics, driven by our desire to understand our respective subjects. By outsourcing our labour to GenAI, we not only adulterate the integrity of academic research but also undermine our own hard-won skills and deprive ourselves of the joy of intellectual pursuits in favour of mediocre expeditiousness.

### Mentorship

In our roles as educators we are responsible for teaching, inspiring and supporting students in their academic lives through conscientious mentorship and a genuine interest for our subjects – facets that cannot be hewn from GenAI.

We fulfil our roles as educators by demonstrating academic rigour and modelling research integrity. Furthermore,

on an intangible level, successful outcomes in students frequently involve meeting the optimism with which students are eager to learn and demonstrate interest and excitement for our subjects. Through these actions, students not only grow in their categorical understanding of a subject but also develop confidence in their academic abilities, thereby achieving their goals. However, by using GenAI to replace our passion and interest for a subject, we are depriving our students and mentees of the opportunity for authentic scholarship. Indeed, all the aspects of successful education, beyond providing the course content, are displaced when GenAI is used to plan, prepare, grade and deliver educational material. Rather than having a positive impact on the students' academic lives it will, at best, be a forgettable experience, at worst, a neglectful one.

In mentorship/supervisory roles the underlying architecture of support remains, however, with an additional layer of deeper personal involvement. An academic mentor metaphorically climbs the mountain with their student, rather than encouraging them from the top. Indeed, this role can be equally educational for the mentor as for the mentee, allowing new avenues of exploration and novel idea generation, as well as being deeply rewarding (Montgomery et al., 2014). Given the tremendous impact that good mentorship can have on students, it is imperative that the appropriate and conscientious use of GenAI tools is modelled by the mentor. Furthermore, the dynamic relationship between student and supervisor cannot be substituted by the inauthentic and detached relationship created by GenAI.

## Education

At higher levels of education, the narrowly defined paths of secondary education open up into a vast landscape of opportunity, with the freedom to pursue reckless scholarship propelled by genuine interest and an eagerness to learn. It is therefore disheartening to witness the ready and blithe adoption of GenAI to complete coursework and assignments by many students.

Although specific learning objectives and teaching outcomes still define the structure of the undergraduate experience, this is the first step on the path to independent scholarship. This path is meant to be inspiring and enjoyable: by encouraging and shaping a fundamental interest in a subject through opportunities to read a broad range of sources, write on diverse subject matters and learn from honest mistakes, higher education seeks to equip students with important scholarly tools, as well as galvanise their passion. In other words, this is an opportunity to learn how to learn and how to engage critically with a subject. For example, assignments are not set to satisfy the needs of the educator – these are chances given to engage with a subject, learn and improve. As such, using GenAI to complete coursework is entirely reductive and only serves to impoverish the educational experience that is being sought.

Indeed, academic education should not be about arriving at a destination with passed exams, completed assignments and a degree certificate. It should be about learning to synthesise new thought through genuine creative reasoning and developing a deeper understanding of a subject in the process.

## Conclusion

Regardless of career stage, the creative and abstract thinking skills that enable us to investigate the world, solve problems and innovate are motivated by a curiosity that is fundamental to academic scholarship. GenAI is not only unable to capture the nuance of organic thought or the sophisticated reasoning of a scholar, but its overabundant and negligent use also robs us of the privilege to learn, read and think.

Given that I personally would not be inclined to relinquish the best parts of my academic job to another person in my stead, I am also not interested in outsourcing these aspects to the probabilistic algorithm of GenAI. As such, my plea is to consider the underlying nature and intention of any assignment, project or, indeed, career, by asking if the final destination is more important than the journey taken to reach it. Rather than taking an irresponsible and reductive shortcut, my counsel is to take a cautious and deliberate approach to GenAI in the name of genuine, enthusiastic academic scholarship.

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

## Additional information

### Competing interests

The authors declare no conflict of interest.

### Author contributions

Daniela Schnitzler: conception or design of the work; drafting the work or revising it critically for important intellectual content; final approval

of the version to be published; agreement to be accountable for all aspects of the work.

## Funding

None.

## Keywords

academia, generative AI, opinion

## Supporting information

Additional supporting information can be found online in the Supporting Information section at the end of the HTML view of the article. Supporting information files available:

**Peer Review History**

