## [Peer Review History · The Journal of Physiology]

Generative AI in Academia: Efficiency vs. Scholarship

Daniela Schnitzler

DOI: 10.1113/JP290275

Corresponding author(s): Daniela Schnitzler (daniela.schnitzler@medicalschooll-berlin.de)

The following individual(s) involved in review of this submission have agreed to reveal their identity: Ali Zifan (Referee #1)

Review Timeline:	Submission Date:	07-Oct-2025
	Editorial Decision:	20-Oct-2025
	Revision Received:	06-Nov-2025
	Accepted:	18-Nov-2025

Senior Editor: Kim Barrett

Reviewing Editor: Vaughan Macefield

Transaction Report:

Dear Dr Schnitzler,

Re: JP-OP-2025-290275 "**Generative AI in Academia: Efficiency vs. Scholarship**" by Daniela Schnitzler

Thank you for submitting your manuscript to The Journal of Physiology. It has been assessed by a Reviewing Editor and by 2 expert referees and we are pleased to tell you that it is acceptable for publication following satisfactory revision.

REVISION CHECKLIST:

We look forward to receiving your revised submission.

Yours sincerely,

Kim Barrett
Senior Editor
The Journal of Physiology

EDITOR COMMENTS

Reviewing Editor:

Thank you for submitting your Opinion piece to The Journal of Physiology. It has been assessed by two independent reviewers who both see merit in your article, but recommend that it needs to be more balanced, including discussion on the positive benefits of AI in academia. I invite you to provide detailed responses to both reviewer's comments and look forward to receiving a revised manuscript in due course.

REFeree COMMENTS

Referee #1:

The paper addresses an important and timely topic, and that is the impact of generative AI on academic integrity, creativity, and scholarship. However, several areas could be improved to make it stronger:

a) Introduction: It's not clear at first that this is an opinion piece. I suggest adding a line like:

In this opinion piece, I reflect as an (early/mid?) career researcher on how GenAI is reshaping (my) academic work and integrity, in the first paragraph.

b) Same Intro section: I suggest briefly adding 1 to 2 sentences on how universities are currently debating or creating policies around AI use and misuse.

c) Research section: The arguments feel repetitive around 'outsourcing thinking' and 'loss of creativity.' Just merge paragraphs 2 to 4 and keep only one strong example, such as using GenAI to summarize papers. Also, remove repeated phrases about critical thinking and reasoning limits.

d) Still in the research section: Right now the tone is quite negative. It would be good to include 1 to 2 sentences on the benefits of GenAI. Just to keep it balanced. For example, how it can help with data sorting, improve accessibility for non-native English speakers, or support early idea development, etc.

e) Continuous Scholarship section: This overlaps with the research section. Delete the first two sentences, since they repeat earlier points. Just focus on learning and intellectual growth. You could mention how GenAI may reduce engagement with diverse sources, rather than general creativity.

f) Mentorship section: This part sounds a bit too emotional, especially words like 'inauthentic' and 'cheating.' I suggest rephrasing to something more professional, for example: Overreliance on GenAI risks weakening authentic mentor-mentee engagement and intellectual development.

You can keep the mountain metaphor if you want. It adds a nice personal touch.

g) Also in the mentorship section: It would be helpful to note that mentors should model ethical and responsible GenAI use, so students can learn from their example.

h) Education section: This section is way too long and repeats earlier points about critical thinking and integrity. You can easily cut it in half. Keep the focus on how GenAI may undermine student learning and self-discovery.

i) Conclusion: It is not clear to the reviewer, the conclusion. Reads vague. I suggest replacing the last two paragraphs with something more clear and forward-looking. Also consider adding a short personal statement or vision of what balanced and ethical GenAI use should look like. Since this is an opinion piece, it would give the ending a stronger and more personal finish.

Referee #2:

The opinion article "Generative AI in Academia: Efficiency vs. Scholarship" by Daniela Schnitzler expresses concerns about the use of artificial intelligence (AI) in academia. While the article raises valid concerns, in my opinion, it is too generalized, accusatory, and needs to be more nuanced. For example, statements such as "... many of the facets intrinsic to academia cannot simply be replaced or expedited through GenAI and to do so is antithetical to academic thought and intellectual pursuits. ", "GenAI-based strategies undermines many of the intellectual pursuits intrinsic to academic research ...", "By outsourcing our labour to GenAI we not only adulterate the integrity of academic research, but we also undermine our own hard-won skills and deprive ourselves of the joy of intellectual pursuits in favour of mediocre expeditiousness." are too superficial, in parts problematic, and disregard the many creative, productive uses of AI in academia. The way I understand this article is that it generally accuses scientists who adopt AI to "rob" themselves of their opportunity to grow, which I vehemently disagree with.

END OF COMMENTS

Dear Editors and Referees,

I thank you for the thoughtful comments and suggested edits to my opinion piece. I appreciate the effort taken to help improve the work and have incorporated the majority of the suggestions made by the reviewers. Please find below the responses to each point and details on their implementation (in red).

Kind Regards,

Daniela Schnitzler

Reviewing Editor:

Thank you for submitting your Opinion piece to The Journal of Physiology. It has been assessed by two independent reviewers who both see merit in your article, but recommend that it needs to be more balanced, including discussion on the positive benefits of AI in academia. I invite you to provide detailed responses to both reviewer's comments and look forward to receiving a revised manuscript in due course.

Thank you - you will find below detailed responses to each comment.

Referee #1:

The paper addresses an important and timely topic, and that is the impact of generative AI on academic integrity, creativity, and scholarship. However, several areas could be improved to make it stronger:

a) Introduction: It's not clear at first that this is an opinion piece. I suggest adding a line like:

In this opinion piece, I reflect as an (early/mid?) career researcher on how GenAI is reshaping (my) academic work and integrity, in the first paragraph.

Added.

b) Same Intro section: I suggest briefly adding 1 to 2 sentences on how universities are currently debating or creating policies around AI use and misuse.

Added.

c) Research section: The arguments feel repetitive around 'outsourcing thinking' and 'loss of creativity.' Just merge paragraphs 2 to 4 and keep only one strong example, such as using GenAI to summarize papers. Also, remove repeated phrases about critical thinking and reasoning limits.

I suspect paragraphs 2 and 3 were meant here, as paragraph 4 questions the use of GenAI beyond its technical limitations. As such, I have removed paragraph 3, but kept the ideas and examples, as they speak to different aspects of use.

d) Still in the research section: Right now the tone is quite negative. It would be good to include 1 to 2 sentences on the benefits of GenAI. Just to keep it balanced. For example, how it can help with data sorting, improve accessibility for non-native English speakers, or support early idea development, etc.

Added.

e) Continuous Scholarship section: This overlaps with the research section. Delete the first two sentences, since they repeat earlier points. Just focus on learning and intellectual growth. You could mention how GenAI may reduce engagement with diverse sources, rather than general creativity.

Moved idea into "Research" section to avoid repetition.

f) Mentorship section: This part sounds a bit too emotional, especially words like 'inauthentic' and 'cheating.' I suggest rephrasing to something more professional, for example: Overreliance on GenAI risks weakening authentic mentor-mentee engagement and intellectual development.

Replaced "cheating" but kept "inauthentic" as it reflects my strong opinion.

You can keep the mountain metaphor if you want. It adds a nice personal touch.

g) Also in the mentorship section: It would be helpful to note that mentors should model ethical and responsible GenAI use, so students can learn from their example.

Added.

h) Education section: This section is way too long and repeats earlier points about critical thinking and integrity. You can easily cut it in half. Keep the focus on how GenAI may undermine student learning and self-discovery.

Removed paragraph about postgrad education since this is the same researchers and eliminates repetition.

i) Conclusion: It is not clear to the reviewer, the conclusion. Reads vague. I suggest replacing the last two paragraphs with something more clear and forward-looking. Also consider adding a short personal statement or vision of what balanced and ethical GenAI use should look like. Since this is an opinion piece, it would give the ending a stronger and more personal finish.

Re-worked conclusion to emphasise vision going forward with a more personal statement.

Referee #2:

The opinion article "Generative AI in Academia: Efficiency vs. Scholarship" by Daniela Schnitzler expresses concerns about the use of artificial intelligence (AI) in academia. While the article raises valid concerns, in my opinion, it is too generalized, accusatory, and needs to be more nuanced. For example, statements such as "... many of the facets intrinsic to academia cannot simply be replaced or expedited through GenAI and to do so is antithetical to academic thought and intellectual pursuits. ", "GenAI-based strategies undermines many of the intellectual pursuits intrinsic to academic research ...", "By outsourcing our labour to GenAI we not only adulterate the integrity of academic research, but we also undermine our own hard-won skills and deprive ourselves of the joy of intellectual pursuits in favour of mediocre expeditiousness." are too superficial, in parts problematic, and disregard the many creative, productive uses of AI in academia. The way I understand this article is that it generally accuses scientists who adopt AI to "rob" themselves of their opportunity to grow, which I vehemently disagree with.

As you can see from my answers to referee 1, I have edited my piece in places to provide more nuance and reference to the literature. We will probably still disagree overall, but the purpose of this piece is to lay out my point of view and hopefully spark a spirited discussion. To avoid confusion, I have also signposted earlier in the introduction that this is an opinion piece.

Re: JP-OP-2025-290275R1 "**Generative AI in Academia: Efficiency vs. Scholarship**" by Daniela Schnitzler

Dear Dr Schnitzler,

We are pleased to tell you that your paper has been accepted for publication in The Journal of Physiology.

Yours sincerely,

Kim Barrett
Senior Editor
The Journal of Physiology

IMPORTANT POINTS TO NOTE FOLLOWING ACCEPTANCE OF YOUR PAPER:

- **IMPORTANT NOTICE ABOUT OPEN ACCESS:** To assist authors whose funding agencies mandate immediate public access to published research findings, The Journal of Physiology allows authors to pay an Open Access (OA) fee to have their papers made freely available immediately on publication.

The Corresponding Author will receive an email from Wiley with details on how to register or log-in to Wiley Authors Services where you will be able to place an order

- If you would like to receive our 'Research Roundup', a monthly newsletter highlighting the cutting-edge research published in The Physiological Society's family of journals (The Journal of Physiology, Experimental Physiology, Physiological Reports, The Journal of Nutritional Physiology, and The Journal of Precision Medicine: Health and Disease), please click this link, fill in your name and email address and select 'Research Roundup': <https://www.physoc.org/journals-and-media/membernews>

- You can help your research get the attention it deserves! Check out Wiley's free Promotion Guide for best-practice recommendations for promoting your work at: www.wileyauthors.com/eoo/guide. You can learn more about Wiley Editing Services which offers professional video, design, and writing services to create shareable video abstracts, infographics, conference posters, lay summaries, and research news stories for your research at: www.wileyauthors.com/eoo/promotion.

EDITOR COMMENTS

Reviewing Editor:

Thank you for addressing the reviewer's concerns, both of whom are satisfied with your amendments.

REFEREE COMMENTS

Referee #1:

The author has addressed the review comments.

Referee #2:

The author has carefully revised their opinion article and adopted a more balanced tone. The article is now less accusatory and judgmental, reading more like a pledge to scientific integrity and a call for cautiousness with AI. While I agree with the general sentiment, I still disagree with the author's negative portrayal of AI use in research. The article still contains some statements which I find problematic, such as "By outsourcing our labour to GenAI we not only adulterate the integrity of academic research, but we also undermine our own hard-won skills and deprive ourselves of the joy of intellectual pursuits in favour of mediocre expeditiousness.", which is unfortunate. However, I respect their opinion, and since this is an opinion article, I can only disagree with certain aspects of their writing.